# On-Skin Flexible Pressure Sensor with High Sensitivity for Portable Pulse Monitoring

**DOI:** 10.3390/mi13091390

**Published:** 2022-08-25

**Authors:** Weihao Zheng, Hongcheng Xu, Meng Wang, Qikai Duan, Yangbo Yuan, Weidong Wang, Libo Gao

**Affiliations:** 1School of Mechano-Electronic Engineering, Xidian University, Xi’an 710071, China; 2School of Automation and Software Engineering, Shanxi University, Taiyuan 030013, China; 3Department of Mechanical and Electrical Engineering, School of Aerospace Engineering, Xiamen University, Xiamen 361102, China

**Keywords:** flexible sensor, pressure, high sensitivity, portable, pulse monitoring

## Abstract

Radial artery pulse pressure contains abundant cardiovascular physiological and pathological information, which plays an important role in clinical diagnosis of traditional Chinese medical science. However, many photoelectric sensors and pressure sensors will lose a large number of waveform features in monitoring pulse, which will make it difficult for doctors to precisely evaluate the patients’ health. In this letter, we proposed an on-skin flexible pressure sensor for monitoring radial artery pulse. The sensor consists of the MXene (Ti_3_C_2_T_x_)-coated nonwoven fabrics (n-WFs) sensitive layer and laser-engraved interdigital copper electrodes. Benefiting from substantially increased conductive paths between fibers and electrodes during normal compression, the sensor obtains high sensitivity (3.187 kPa^−1^), fast response time (15 ms), low detection limit (11.1 Pa), and long-term durability (20,000 cycles). Furthermore, a flexible processing circuit was connected with the sensor mounted on wrist radial artery, achieving wirelessly precise monitoring of the pulse on smart phones in real time. Compared with the commercial flexible pressure sensor, our sensor successfully captures weak systolic peak precisely, showing its great clinical potential and commercial value.

## 1. Introduction

Radial arterial pulse pressure (RAPP) monitoring is crucial for Chinese medical-aided diagnosis since rich physiological features in pulse waves can help doctors to evaluate and diagnose diseases within the body, instead of scanning devices [1,2,3,4]. The light wave difference caused by a light–absorption change in human hemoglobin is used to monitor pulse wave by ordinary photoelectric pulse sensors, which usually lose substantial information of RAPP owing to their instability during near-infrared light transmittance [5,6]. In addition, monitoring weak entire RAPP remains challenging for most reported flexible pressure sensors due to their low sensitivity [7,8]. Hence, it is necessary to develop a highly sensitive pressure sensor to precisely monitor arterial pulse.

Currently, most reported works focused on the novel two-dimensional (2D) material used to being sensitive elements in flexible pressure sensors, such as graphene, carbon nano-tube, and SnSe_2_, etc. [9]. For example, Jing et al. proposed a flexible piezoresistive pressure sensors based on graphene, which achieved high elasticity of 85% but exhibited a limited sensitivity (0.075 kPa^−1^) and long response time (120 ms) [10]. Park et al. proposed a flexible sensor based on carbon nanotube thin films, which exhibited a high sensitivity of 278.5 kPa^−1^ but a limited responding range of 0–30 Pa [11]. In addition, the sensor based on SnSe_2_ nanosheet reported by Tannarana et al. can only work under a pressure beyond 2 kPa, resulting in disability for monitoring RAPP [12]. MXene is intrinsically hydrophilic and yet they have demonstrated higher electrical conductivity than solution processed graphene [13], thereby enhancing sensor’s sensitivity greatly by the reduction of contact resistance and possibly enabling sensors to attain more pulse. In addition, their exceptional electrochemical properties were widely used in flexible sensors [14]. Consequently, MXene-based flexible pressure sensor (FPS) is a potential candidate to solving problems like low sensitivity, long response time, and a small working range of forementioned sensors.

Here, an integrated FPS composed of laser-engraved interdigital copper electrodes and the MXene/n-WFs (nonwoven fabrics) sensing layer was proposed. A key sensitive element was made of MXene-coated n-WFs. The sensor was connected to an on-skin wirelessly flexible processing circuit to successfully monitor RAPP on a smart phone. Owing to substantially increased conductive paths between fibers and electrodes during compression, the FPS achieved a high sensitivity of 3.187 kPa^−1^ within 6 kPa and 1.059 kPa^−1^ in a larger pressure region of 100 kPa, as well as a low-pressure detected limit down to 11.1 Pa. Furthermore, FPS is able to monitor RAPP precisely compared with a commercial pressure sensor and can be used to help for diagnosis of cardiovascular diseases.

## 2. Experimental Details

### 2.1. Structure Design and Mechanism

During arterial constriction and relaxation, a weak blood pressure was produced by the process periodically forming a pulse waveform (Figure 1a). By converting this mechanical force into an electrical output, a thin and flexible pressure sensor is applied to the wrist skin to capture RAPP in real time and enable long-term monitoring on a smartphone via a back-end processing circuit on the skin (Figure 1b). As shown in Figure 1c, the designed FPS is applied to the wrist skin with the processing circuit. To connect closely with the FPS and meet the wearable needs, we designed the signal processing circuit to be flexible The FPS mainly consists of a PU encapsulation layer at the top, an MXene/n-WFs sensitive layer, interdigital electrodes, and a supporting polyimide (PI) layer (Figure 1d). Among them, the interdigital electrodes are designed to sense local weak signals to improve the sensitivity of the sensor on uneven skin. The two pieces of 3M tape (3M3300LSE-9495LSE, Linxing Company, Shenzhen, Guangdong province, China) are the key to improving the sensitivity of FPS, which is used to increase the initial compression space of the sensitive layer, leading to a high initial contact resistance and the ability of detecting a weak pressure signal. The MXene/n-WFs become contacted with each other when the sensor is squeezed by a weak pulse force, thereby reducing the contact resistance between the fibers. In addition, another reduction in contact resistance is attributed to the contact between sensitive fibers and electrodes under compression. Both of these reasons are formative mechanisms for the sensor to capture external stimuli.

### 2.2. Materials and Fabrication

The material of a sensitive layer determines the property of the sensor. Ti_3_C_2_T_x_ MXene is one of the 2D transition metal carbides/nitrides that possess good metallic conductivity. Figure 2a shows the scanning electron microscope (SEM) image of Ti_3_C_2_T_x_ nanosheet (5 mg·mL^−1^, Xiyan New Materials Company, Nanjing, Jiangsu province China) in which the insert exhibits a Tyndall effect in solution. Because of the existence of a large number of hydroxyl and fluorine groups on the surface, Ti_3_C_2_T_x_ nanosheets can be evenly distributed in solution, which also shows that the metallic conductivity and rich surface functionalities can coexist without mutual interference [15,16]. Benefiting from the irregular microfiber structure and high porosity of n-WFs (250 mm ∗ 380 mm ∗ 500 pieces, Order Flagship Store), MXene can be dipped and coated on the fabric surface as the sensitive layer. To obtain the sensitive layer, the n-WFs were first washed with ethanol and deionized water and dried. Next, the dried n-WFs were immersed in aqueous Ti_3_C_2_T_x_ solution (5 mg mL^−1^) for one minute and dried at 50 °C, and this operation was repeated several times. Figure 2b shows the morphology (inserts) and square resistance of the n-WFs coated for 3, 5, 7, and 9 times. The square resistance decreased from 14.8 Ω cm^−2^ to 5.2 Ω cm^−2^ as the coating number increased. However, the Ti_3_C_2_T_x_ nanosheets on the fibers were severely aggregated when the coating reached nine times, so we used MXene/n-WFs modified by seven times as the sensitive layer in the following study. The fabricated MXene/n-WFs samples showed good mechanical stability in the uniaxial compression test (Figure 2c). When the compressive strain reached 80%, the sensitive layer still exhibited almost uniform loading and offloading processes, which was attributed to the mechanical stability of n-WFs. Compared to pure n-WFs, MXene/n-WFs exhibited almost the same tensile fracture strength of 75 MPa, indicating that the modification of MXene had no effect on its mechanical behavior (Figure 2d). Figure 2e illustrates the fabrication process of the FPS. The interdigital electrodes were engraved on PI films using a laser ablation. The sensitive layer was aligned with the interdigital electrodes, and the prepared sensor was packed in a PU tape on PET film.

## 3. Results and Discussion

### 3.1. Test Platform

The test platform is shown in Figure 3a, which comprises a force gauge (Zhiqu, ZQ-990B, Zhiqv Company, Dongguan, Guangdong province, China) and an electrochemical workstation (CHI760E, Chenhua Company, Shanghai, China) and their respective upper computers. The fabricated sensor was placed on the platform of the force gauge with its electrodes connected to the electrochemical workstation. By applying different pressure to FPS using force gauge, the performance of the sensor is characterized through the variation of current recorded by electrochemical workstation under 1 V voltage.

### 3.2. Device Characterization

Figure 3b shows the normalized current variation of the sensors under different pressures. The FPS shows a high sensitivity of 3.187 kPa^−1^ within 6 kPa and 1.059 kPa^−1^ in a larger pressure region up to 100 kPa, where the sensitivity is calculated by the formula:(1)$$S=\frac{\Delta I/I_{0}}{\Delta P}$$
where the $I_{0}$ is the initial current, ΔI is the variation of current under different pressure, and ∆*P* is the amount of the pressure change from $I_{0}$ to I. It can be seen that the FPS has a good linearity and sensitivity in a working range, especially under the pressure lower than 6 kPa, which performs the advantage of detecting weak pressure. Compared to compression under a larger force, the sensitive layer can form a larger compressed strain at a low pressure range, which is determined by the compression response as shown in Figure 2c. Larger strain in the modified n-WFs can excite more contact between inner conductive fibers to achieve higher resistance decrease, thus to form higher sensitivity at low pressure. Compared to previous works, our sensor still exhibits eminent sensitivity and working range as shown in Figure 3c [17,18,19,20,21,22]. The I–V curves of the FPS show a good ohmic contact between sensitive layer and electrodes that the voltage ranges from −1 to 1 V (Figure 3d). Figure 3e shows a small hysteresis between current change and applied pressure, verifying the good response capability of the sensor to the applied pressure. In addition, the sensor exhibits fast response and recovery times of 15 ms and 36 ms, respectively, at a continuous pressure of 10 kPa (Figure 4a). In addition, our sensor can sense weak pressures down to 11.1 Pa, as shown in Figure 4b, demonstrating the potential of the sensor for weak RAPP monitoring. To evaluate the recoverability of the sensor, the FPS remains stable for 120 s under different continuous forces and recovers immediately after offloading, as shown in Figure 4c. It is important to verify the response of FPS to the force under different curvature because the curvature of the skin at radial arteries of people are different. We attach the FPS to cylinders with different radii and apply different pressure to it, and the results are shown in Figure 4d. It can be seen that the response of sensors with different curvature is approximately the same under the same pressure, demonstrating the ability of the detection of RAPP of different people. In addition, the FPS maintains a constant output without significant signal degradation after 6000 cycles under loading-unloading pressure of 1 kPa (Figure 4d), verifying the stability of FPS to monitor RAPP. Furthermore, we apply a much bigger pressure of 50 kPa to FPS for more than 14,000 cycles, and the output is still stable, showing good durability.

### 3.3. Applications

The fabricated sensor has high sensitivity and linearity, leading to a wide range of applications. As shown in Figure 5a, a balloon is used for a blowing test, and the results show that the FPS is able to monitor human breath. The responses caused by finger touch also prove the potential application of FPS in robotic tactile sensation in complex environments (Figure 5b). As shown in Figure 5c, we fix the sensor to the second joint of the index finger with medical tape and bend the joint into 30°, 60°, and 90° three times, respectively. The increasing current variation proves the ability of FPS to monitor finger movements, so it can be used to collect signals for human finger rehabilitation training. Furthermore, by placing the sensor on the human carotid artery, we successfully collected the human carotid pulse and swallow signals, demonstrating the application of the monitoring of human larynx health.

Considering the significance of RAPP in medical diagnosis and the performance of the FPS, we apply the sensor to long-term remote wireless monitoring of RAPP. To achieve real-time monitoring, a flexible wireless signal processing circuit was designed as shown in Figure 6a. A Wheatstone bridge is used to convert the resistance change of the FPS into a voltage change, and then the voltage signal is transmitted to an amplifier. Next, the analog voltage signal is converted to a digital signal by an Analog to Digital Converter (ADC) and filtered by digital software in the Microcontroller Units (MCU). Finally, the processed signal is sent to the cell phone via a low-power Bluetooth module for remote data transmission and visualization. As shown in Figure 6b, the fabricated thin FPS can be tightly attached to the volunteer’s wrist skin and integrated with the flexible circuit. The real-time monitoring of the RAPP on a smartphone further demonstrates the feasibility of our sensor system in physical scenarios.

Stable and consistent waveform can reflect a health condition of the tested subject. Clinically, pulse waveform is a key indicator of many cardiovascular diseases, and it mainly includes a systolic peak ($P_{1}$), a reflected systolic peak ($P_{2}$), a dicrotic notch ($P_{3}$), a diastolic peak ($P_{4}$), and a diastolic valley ($P_{5}$) [23,24]. These peaks can be analyzed and processed to determine the cardiovascular health patients. To evaluate the superiority of our FPS, we compared it with a commercial pulse sensor (RP-C7.6-ST-LF2, brought from LEGACT). During the test, we fix the FPS and commercial sensor with medical tape to a 25-year-old male volunteer’s radial artery successively. It is important to reduce the impact of external environment on the results. Therefore, in the process of fixing the sensors, we try to ensure the consistent position of two sensors, and let the tape exert a pressure of about 1 kPa on the sensors to reduce the impact of external pressure. The results obtained are normalized and compared, and the RAPP peaks are marked, as shown in Figure 7a. It can be seen that there are obvious differences of the two pulse waveforms. The values of $P_{2}$ and $P_{5}$ of two curves are basically the same, but the commercial sensor hardly detects the characteristics of $P_{2}$ peak, which leads to the difference of $P_{3}$ peak compared to FPS. Hence, compared to a commercial sensor, our FPS can keep more characteristics of $P_{2}$ and $P_{3}$. The values of $P_{2}$ and $P_{3}$ can be used to evaluate important physiological and pathological indexes of the human body, such as the radial artery augmentation index ($AI_{r}=P_{2}/P_{1}$) and the radial diastolic augmentation ($DAI=P_{3}/P_{1}$) [25,26]. Hence, the RAPP waveform monitored by FPS can be observed to predict diseases such as arteriosclerosis and hypertension. Furthermore, according to the pulse wave theory [25,27,28], the mean arterial pressure $P_{m}$ in the radial artery is calculated as follows:(2)$$P_{m}=\frac{1}{T}\int_{0}^{T}p\left(t\right)dt=P_{5}+K\left(P_{1}-P_{5}\right)$$
where $p\left(t\right)$ is the value of RAPP waveform, K is the waveform coefficient of RAPP and a parameter indicating the degree of arteriosclerosis, and we can obtain that:(3)$$K=\frac{P_{m}-P_{5}}{P_{1}-P_{5}}$$

Consequently, the K is just related to the shape of pulse pressure waveform [27]. It corresponds to the ratio of the mean value and the peak value of RAPP. Vascular resistance and arterial elasticity of healthy young people are low with a K value of about 0.32–0.39. The middle-aged and elderly people have higher K value, which is about 0.4 owing to blood viscosity increase. The K value of patients with severe hypertension and atherosclerosis is about 0.45–0.5. The K (0.344) we obtained from our sensor reflects the healthy cardiovascular status of an adult male. The RAPP waveform can also be used to monitor the heart rate. Figure 7b shows the spectrum of the monitored RAPP waveform, where the volunteer’s heart rate is 1.3 Hz. Furthermore, we monitored the RAPP for a period of time, and the waveform remained stable, indicating the stability of FPS (Figure 7c). Figure 7d shows the short-time Fourier transform (STFT) signals of Figure 7c. Judging from the frequency component at each time, we can obtain the human cardiovascular activity.

## 4. Conclusions

In this work, an on-skin flexible pulse pressure sensor was proposed and connected with a flexible processing circuit to successfully monitor radial artery pulse wirelessly. MXene-modified n-WFs as the sensing layer markedly improve the sensitivity compared to many previous works. Furthermore, a flexible functional circuit was connected with the sensor and provides an epidermally real-time and wireless monitoring for pulse wave on the wrist. The sensor also can distinguish clear pulse wave peaks related to a commercial sensor. By rationally building sensing material into a thin sensor and integrating with an on-skin system instead of merely developing sensors, we hope this design will facilitate the development of portable clinics in the future.

## Figures and Tables

**Figure 1 micromachines-13-01390-f001:**
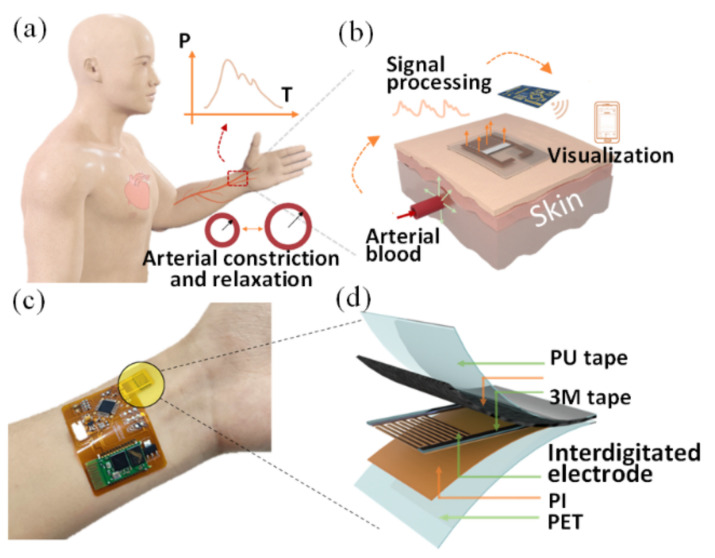
(**a**) The beating of the human heart causes the radial artery vasodilatation and vasodilatation, resulting in a slight pressure change on the skin surface; (**b**) collecting the pressure response signals by flexible sensor and sending signals to the mobile phone for display by a signal processing circuit; (**c**) optical image of the FPS connected with a processing circuit mounted on a participant’s wrist; (**d**) schematic diagram of the structure of FPS.

**Figure 2 micromachines-13-01390-f002:**
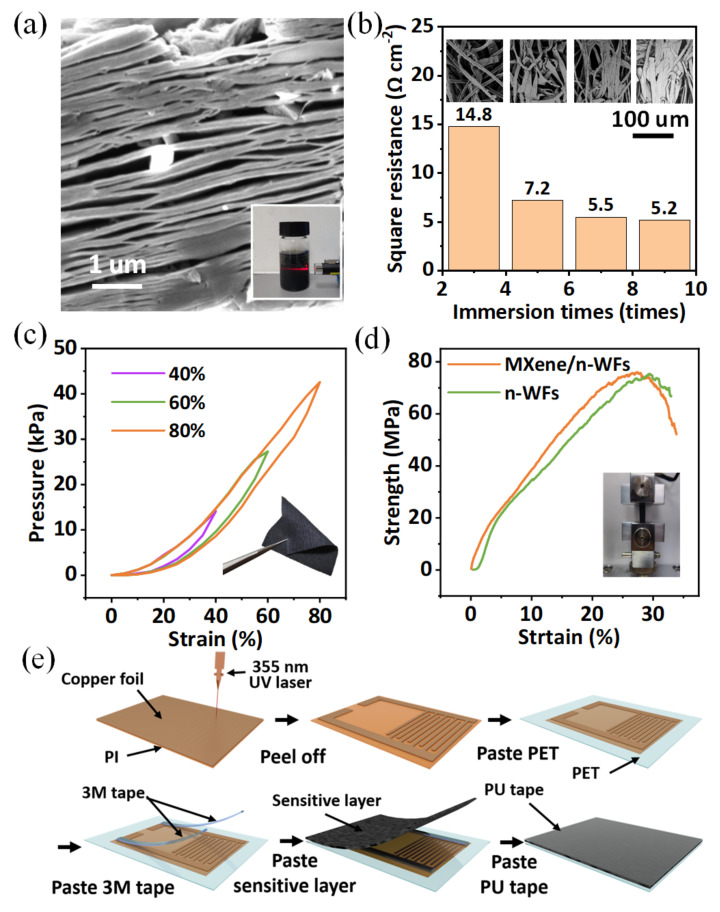
(**a**) SEM image of Ti_3_C_2_T_x_ nanosheet and Tyndall effect of Ti_3_C_2_T_x_ nanosheets solution; (**b**) morphology and square resistance of the n-WFs coated for 3, 5, 7, and 9 times; (**c**) compression test’s comparison under various strains; (**d**) tensile test of the n-WFs and MXene/n-WFs; (**e**) fabricated process of FPS.

**Figure 3 micromachines-13-01390-f003:**
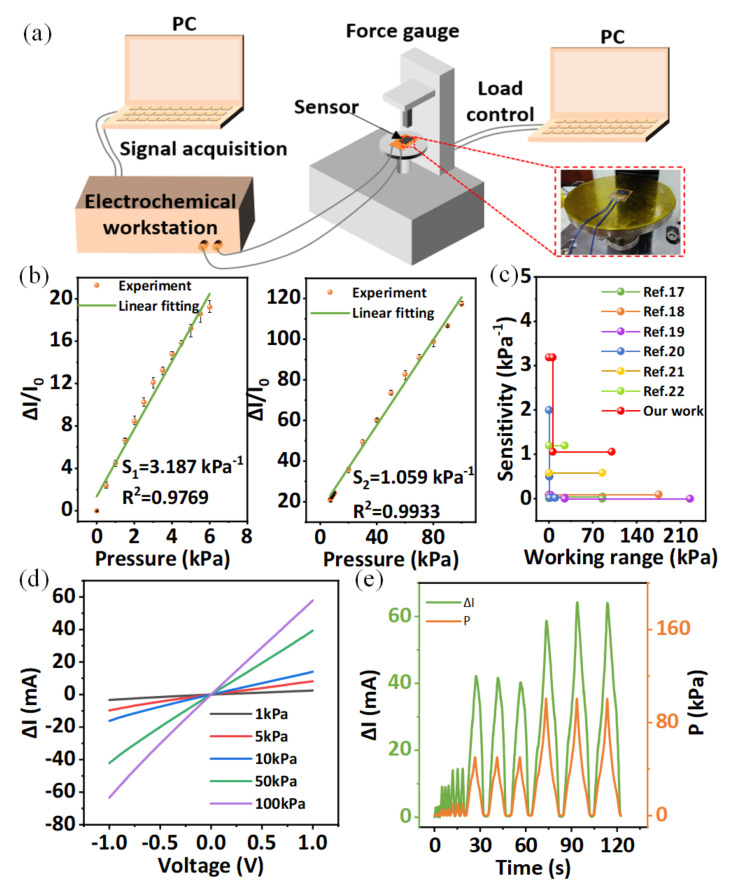
(**a**) Schematic illustration of the experimental setup; (**b**) the sensitivity of FPS, showing a high sensitivity of 3.187 kPa^−1^ within 6 kPa and 1.059 kPa^−1^ in a larger pressure region up to 100 kPa; (**c**) comparison of the fabricated sensor with other flexible sensors; (**d**) I–V curves of the sensor under various applied pressures; (**e**) current response under increasing pressures.

**Figure 4 micromachines-13-01390-f004:**
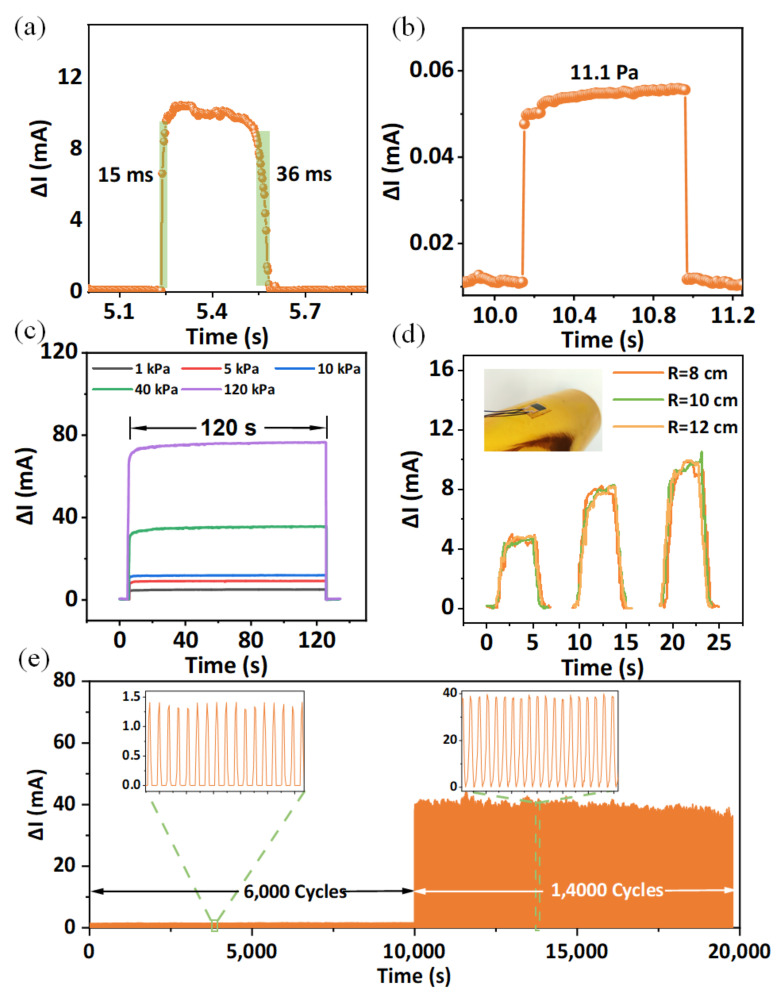
(**a**) Outlined response and relaxation time under a pressure of 10 kPa; (**b**) minimum detection pressure is 11.1 Pa; (**c**) response of FPS under different pressure for 120 s; (**d**) The FPS were tested on the sides of cylinders with different radii; (**e**) long-term durability over 6000 cycles under a pressure of 1 kPa and 14,000 cycles under a pressure of 50 kPa.

**Figure 5 micromachines-13-01390-f005:**
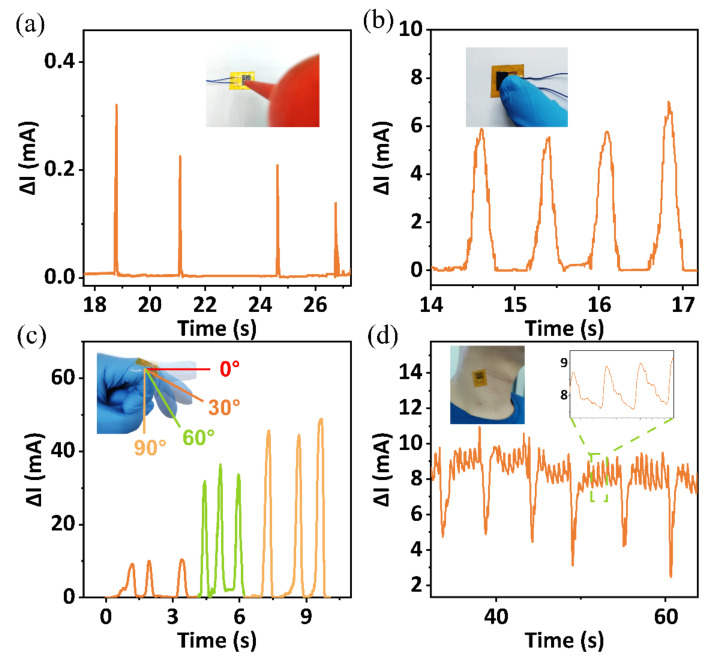
(**a**) The current response output of gas leak monitoring, and the image is shown in the inset; (**b**) the response of touching; (**c**) the response of bending the finger; (**d**) the monitoring of swallowing and carotid pulse.

**Figure 6 micromachines-13-01390-f006:**
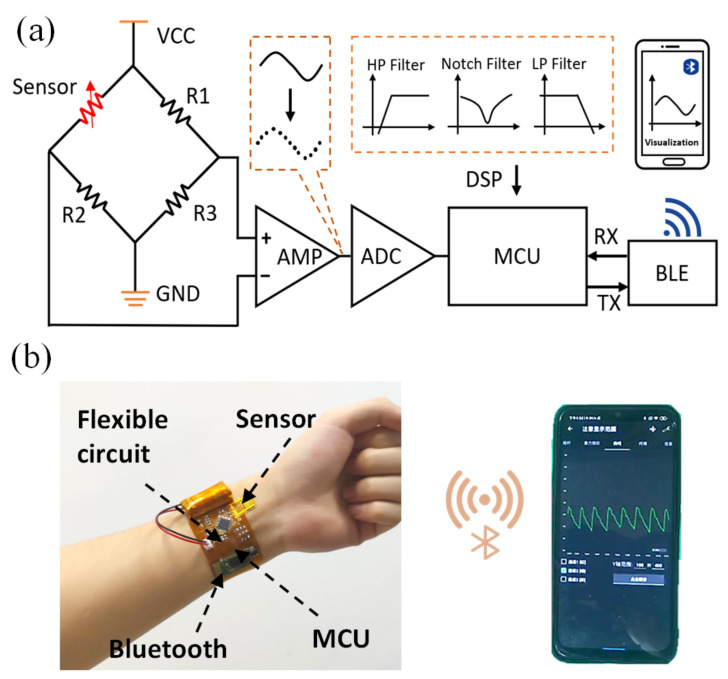
(**a**) Functional block diagram of pulse pressure signal acquisition; (**b**) The FPS is attached to the skin together with the flexible processing circuit. By connecting to the Bluetooth of mobile phone, the remote monitoring of RAPP was achieved.

**Figure 7 micromachines-13-01390-f007:**
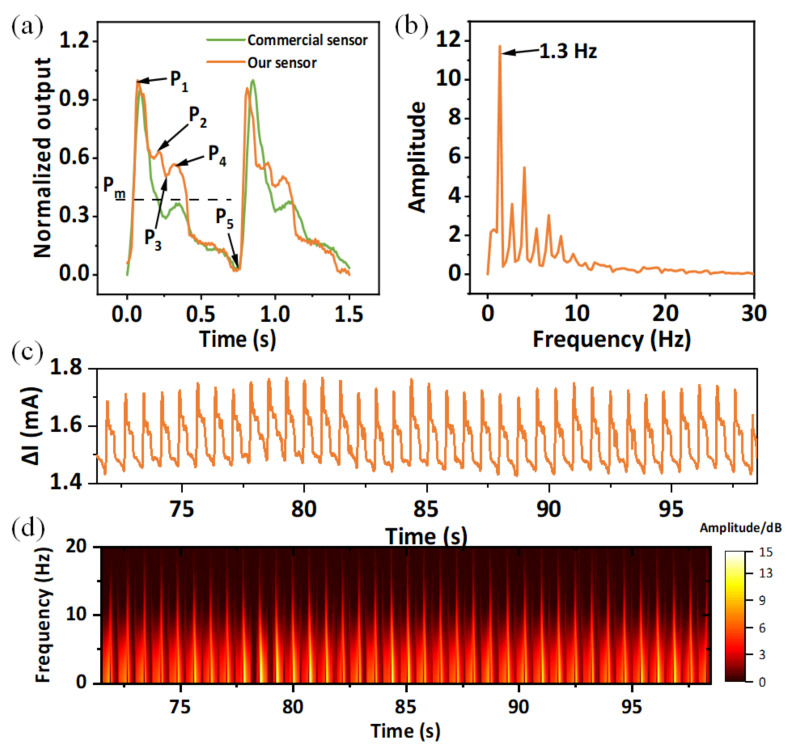
(**a**) Analysis and of RAPP waveform; (**b**) spectrum of (**a**); (**c**) long-term RAPP monitoring; (**d**) the STFE image of (**c**).

## Data Availability

The data presented in this study are available on request from the corresponding author.

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
