# Peer review of "On-Skin Flexible Pressure Sensor with High Sensitivity for Portable Pulse Monitoring"

_micromachines, 2022, doi:10.3390/mi13091390_

Round 1
Reviewer 1 Report
Recommendation: Minor revision before acceptance.
Comments: In this manuscript, the author developed an on-skin flexible pressure sensor for portable pulse monitoring. The whole system can monitor clear pulse signal and show good performance. Overall, the research is interesting and the manuscript is well written. I recommend publishing the work if the following issues are well addressed:
Comment 1: In Figure 1c, the sensor and flexible circuit are not sticky, how to attach them to the skin surface?
Comment 2: In Figure 3b, why the sensor has different sensitivity in different pressure range?
Comment 3: In Figure 4b, how the low detection limit of 0.0005N is measured?
Comment 4: In Figures 7d, what is the purpose of the short-time Fourier transform?
Reviewer 2 Report
This manuscript reports an on-skin flexible pressure sensor based on the MXene‐coated nonwoven fabrics (n‐WFs) sensitive layer and interdigital copper electrodes. The developed sensor has been compared with a commercial pulse sensor on radial artery pulse pressure detection. It is promising that the developed sensor is able to detect more details of the pulse pressure. This manuscript can be accepted by addressing the following issues/comments:
1. The English language of the manuscript needs to be polished. A few examples:
1 Line 38: in “Currently, most reported works just focused on the novel … “, it is better to delete just
2 Line 51: it is better to replace forementioned with aforementioned.
2. Please provide dimensions of the sensor and sensor components.
3. Please provide more information of the n-WFs and MXene, such as vendor information, part number, etc.
4. Line 96: please double check the unit of square resistance.
5. Please indicate the type of 3M tape.
5. From Figure 2(e), it looks like the 3M tape was only attached to the two big side metal traces, not the fine interdigitated electrodes. If this is true, the sensing layer is slightly suspended above the interdigitated electrodes. How is the device-to-device variation for this kind of contact? It is also expected that the sensor's performance is sensitive to the way the sensor is attached to the skin (e.g., the strength of the tape). Please comment on this issue.
